# Pen+Touch+Midair: Cross-Space Hybrid Bimanual Interaction on Horizontal Surfaces in Virtual Reality

Fabrice Matulic*  
Preferred Networks Inc.

Daniel Vogel†  
University of Waterloo

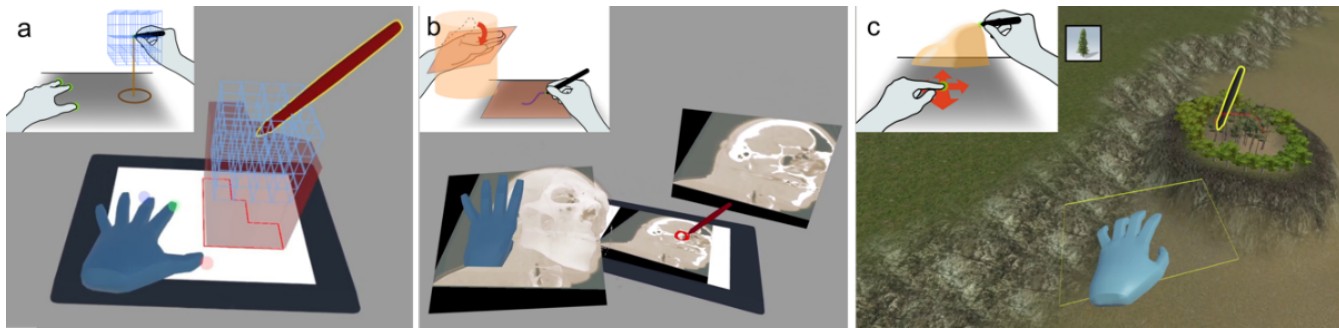

Figure 1: Examples using a pen+touch+midair design space for integrating horizontal touch surfaces into VR: (a) 3-finger touch to activate a grid modelling mode + pen transitioning from 2D surface interaction to midair to extrude a 3D mesh; (b) midair flat hand posture to slice volumetric data for pen annotations on the surface; (c) midair terrain editing with the pen while moving the camera with touch on the surface.

## ABSTRACT

We present and explore a design space for hybrid bimanual pen and touch input extended to midair interaction in desktop-based virtual reality (VR). The investigation focuses on asymmetric interaction patterns combining the pen with the other hand when interacting in the same "space" (either surface or midair), across both spaces, and with cross-space transitions (from surface to midair and vice versa). To show how these interactions and associated gestures can work in context, we create three testbed applications for modelling, volumetric rendering, and terrain editing. A qualitative evaluation with 16 participants provides insights into hand and space preferences for key tasks including object manipulation, navigation, and menu invocation. From the results, design implications are identified for VR systems that combine pen, touch, and midair input, and extensions to other forms of extended reality are discussed.

**Keywords**: pen, touch, bimanual interaction, hand gestures, VR

**Index Terms**: Human-centered computing~Human computer interaction (HCI)~Interaction paradigms~Virtual reality; Human-centered computing~Human computer interaction (HCI)~Interaction devices~Touch screens; Human-centered computing~Human computer interaction (HCI)~Interaction techniques

## 1 INTRODUCTION

Pen and touch input has been investigated in VR [12,19,50,63,73] and AR [3,64,84] (collectively referred to as "XR"), but primarily with handheld devices such as tablets and phones. A handheld device can be used in XR both for its direct input capabilities and as a manipulable object like a slate or wand: for example, drawing

---

*e-mail: fmatulic@preferred.jp  
†e-mail: dvogel@uwaterloo.ca

with a pen in midair [20,78], or one hand positioning a handheld tablet in 3D space while the other hand interacts on it using touch or pen input [3,19,73]. Essentially, a phone, tablet, or pen functions like a special kind of XR controller.

Conversely, large desktop graphics tablets cannot be manipulated in space, but they leave both hands free to interact. A particularly compelling form of interaction with these types of horizonal surfaces is an asymmetric or hybrid combination of pen and touch using both hands [25]. This has been extensively investigated outside of XR, with numerous examples of bimanual techniques, interaction vocabularies and applications having been proposed [13,30,36,52,61,79,83]. Some non-XR systems have added midair input above the tablet surface as well [5,7,49], but they still rely on a horizontal 2D display. In XR, tracking hands or pens in 3D space is required to interact, with the 3D XR display allowing visual feedback for the midair input to be rendered in place. This enables potentially powerful combinations of constrained but precise surface-bound 2D interaction with unconstrained midair 3D input [4,19]. Yet, it remains unclear how different modal and spatial combinations of pen and touch interactions can be used on and above an interactive surface in XR. Systems exploiting the capabilities of the pen-holding hand and the other hand when one is in mid-air and the other is contacting the surface remain scarce and examples of simultaneous two-hand cross-space input almost non-existent.

We contribute the definition and exploration of a design space for bimanual pen and barehand input on and above a horizontal desk-bound interactive surface in XR. We use VR to build different kinds of interactions, but we hypothesise our framework can generalise to other forms of XR. Our exploration is grounded in asymmetric bimanual interaction in the context of pen and touch input and the "continuous interaction space" on and above digital surfaces [48]. We identify combinations and transitions across modalities (pen and bare hand) as well as across physical spaces (surface contact and midair). We provide a high-level overview of such cross-space interactions through several example gestures within three different application contexts: extrusion-based modelling, annotating of volumetric data, and terrain editing (Figure 1). We evaluate these interactions in a qualitative user study with 16

participants. Based on observations and participant feedback, we analyse the strengths and weaknesses of composite interactions and propose research directions for further experimental studies. Our goal is to inform the design of future applications fusing pen and barehand input with interactive surfaces and spaces in VR, with likely applications to XR in general[1].

## 2 RELATED WORK

We review work that combines pen, touch, and midair input with interactive surfaces in various ways. After a brief mention of non-XR systems, we focus on XR.

### 2.1 Pen, Touch, and Midair Input with non-XR Surfaces

Prior work has looked at augmenting the capabilities of pen and touch input with above and around-surface interaction. For example, using the pen in midair to point at out-of-reach targets on a large tabletop [7], arranging virtual flowers [82], and triggering pen mode switches with gestures of the other hand beyond the small screen space of a tablet [5,49]. Concrete examples of the continuous interaction space with transitions between touch and midair input, have also been proposed on mobile devices [15,31,35]. These systems and techniques were developed for 2D displays with all interactive feedback and output confined to a flat 2D space. However, the systematic approach in those works and the breadth of proposed gestures and applications provide a solid foundation for extending these techniques into a XR context, where objects can be directly manipulated in both 2D and 3D spaces.

### 2.2 Asymmetric Bimanual Input with Hands and Controllers

In VR, tracked controllers with buttons commonly provide the basic means to interact, but more direct, embodied experiences support barehand input. In either case, physical metaphors based on symmetric or asymmetric two-hand interaction have been proposed. For instance, following a handle bar metaphor, users can "carry" virtual objects with both hands together or perform axis-constrained rotations with one hand fixing the axis and the other performing a crank-turning motion [69]. Another example is mimicking writing on the palm, where one hand is used as a physical device-like surface for sketching with a finger of the other hand [38]. Analogous to surface pen and touch, the other hand can anchor a tool menu for the dominant hand [38,45,53], directly set dominant hand modes via different hand postures [43,74], or define guides and constraints for precise object positioning and manipulation with the dominant hand [33].

If a bare hand is paired with a controller, interaction can reflect the nature of the different input instruments. Asymmetric roles can be assigned based on associated degrees of freedom (DoF) and input constraints. For instance, a 6 DoF controller can control a fishing rod while the other hand spins the reel [37]. Furthermore, as shown by studies on midair hand input for wall displays [59], object transformation operations can also be broken down and assigned to different hands for increased control [6]. While these works do not use pens, they highlight the expressive power of asymmetric bimanual input. We touch on issues of DoF and control afforded by one or two hands with navigation gestures and marking menu selection techniques in our terrain editing application.

### 2.3 Solo Midair Pen Input

Pens have certain advantages for XR input. In a comparison between controller, mouse, and pen for pointing, Pham and Stuerzlinger found the pen most effective and preferred [62]. Li et al. further investigate pens as pointing devices when adopting different grips to increase raycasting precision and switch modes [44]. Jackson et al. propose midair text input techniques using a pen and find that, while they are slower than text entry based on controller pointing, one of the pen techniques yields fewer errors and causes less fatigue. Drawing on Air is a bimanual technique combining a haptic-aided pen and barehand input to tape-draw 3D curves with the other hand controlling the tangent along which the pen moves [39]. We incorporate solo pen midair input into our design space, and extend this fundamental interaction with variations such as exploring different triggers, transitioning from the touch surface to midair, and using the top end of the pen for midair input.

### 2.4 Pen or Touch with 3D-Tracked Mobile Devices

Introducing a mobile device capable of sensing touch or pen input on a hard surface into a 3D-tracked XR space provides physical support for 2D sketching and tracing with a pen or a finger. Arora et al. have shown this results in higher precision compared to barehand midair input [4]. Using pen-based tablets and mobile pads in XR was first explored decades ago with systems like Personal Interaction Panel [75], Virtual Notepad [63], Transparent Props for the Virtual Table [65], and others [12,21,28,46]. One of the most comprehensive early platforms was Studierstube, a collaborative AR framework for tablet+pen interaction. Example applications include using the tablet as a physical prop to define planes and using the pen to use tools such as a magic lens [66].

The ability of the mobile device to serve not only as a physical surface for precise sketching but as a 6 DoF slate tracked in space was exploited by several later systems. For instance, mobile phones can be used as handheld controllers to point at virtual objects, with tools to manipulate them selected using the touchscreen [57,85]. Phones can also be combined with a bare hand or a regular controller to support asymmetric bimanual interaction, e.g. to separate pointing and touch input roles [50,56]. ARPen is another system that uses a phone to both track a pen using the rear camera and view the augmented content on the screen [78]. The thumb of the hand holding the device is also used to select objects on the display, which is an example of mobile asymmetric bimanual input.

Tablets have a larger surface, which makes them even more suitable for augmented sketching. Napkin Sketch shows sketches drawn on a tablet in AR over a "napkin" physical surface that can be turned to view the sketch from different angles [81]. SymbiosisSketch [3] and VRSketchIn [19] are two pen-based VR systems that allow the user to define virtual surfaces in 3D space mapped to the tablet display for precise surface-bound sketching. They also support unconstrained midair input for freehand sketching in the VR space. VRSketchIn further defines a pen+tablet design space focused on sketching techniques and operations. Slicing Volume is a related selection technique, which allows the user to filter sections of point clouds via bounding volumes and project the enclosed elements onto the tablet for inspection and precise selection with the pen [58]. TabletInVR is a touch-based tablet system with a set of multitouch gestures to perform operations on 3D cubes, where the physical position and orientation of the tablet can also be used to define extrusion directions and to cut through virtual volumes [73].

Mobile devices used in midair require a hand to hold them. Our setting, a fixed horizontal surface, such as a digital tabletop or a tablet placed on a desk with no other handheld instrument than the pen, is different, as it allows both hands to participate directly in 2D and 3D input. However, the ability of the pen-holding hand to transition between surface-constrained to unconstrained midair interaction is similar. We take into consideration relevant techniques in these mobile contexts when creating our pen-driven interactions.

---

[1] This work is partially based on our CHI 2022 Late-Breaking Work "Terrain Modelling with a Pen & Touch Tablet and Mid-Air Gestures in VR".

## 2.5 Desk-Based XR with Asymmetric Bimanual Input

XR systems for large fixed horizontal surfaces sacrifice mobility for the benefit of enhanced freehand interaction with tactile feedback from the interactive tabletop or tablet. The space above the surface is available for visualisation of 3D content as well as midair input using both hands.

A common system for desk-based XR is a stereoscopic tabletop display that renders objects in 3D with the aid of user-worn shutter glasses. An early system of that kind is the Responsive Workbench, which supports bimanual hand and stylus manipulations to conduct various interactive scientific visualisation tasks [41]. Symmetric and asymmetric bimanual midair manipulations are used to translate, rotate, and scale objects displayed on the surface. The system does not use touch input and therefore cross-space interactions are not supported. A similar prototype, ErgoDesk, uses trackballs and 3D trackers operated by the other hand to zoom and orient the camera view for pen tasks [22]. Mockup Builder is a later stereoscopic tabletop system for 3D modelling with asymmetric bimanual touch and midair interactions [2]. Sketching and extruding shapes are performed with the dominant hand while the other hand invokes menus, moves objects and sets constraints for extrusions. The stereoscopic nature of the system and the fact that only the index finger and the thumb of both hands are tracked using retracting cables attached to the fingers limit the possibilities of midair interactions. Nevertheless, many of Mockup Builder's dominant-hand gestures could conceivably be performed with a pen and the extrusion techniques in our modelling application are inspired by this system. GeoCake is another stereoscopic tabletop system for geomodelling, which supports multitouch on the surface and midair input with a wand [47]. While cross-space bimanual input is technically possible, the application does not include any hybrid interactions and midair barehand input is not supported. SpaceTop is an AR desktop using a fixed transparent display above the hands [42], where bimanual keyboard and touch-based operations are supported, as well as midair gestures to manipulate virtual objects above the surface, but similarly the gestures are mostly independent or sequential.

The literature also includes desk-based XR environments that use bimanual pen and touch input. DesignAR is an object design workstation, with modelling mainly performed using unimanual pen or touch operations and the 3D AR space above the display primarily used for the visualisation of 3D objects and data [64]. Gesslein et al. developed a spreadsheet application for pen-based tablets in VR, where the augmented space above and around the device can show data in 3D and layered menus [25]. The application is very pen-centric and the other hand is only used to sporadically perform air clicks with the index finger. PoVRPoint is a VR tool to create presentations using three modalities: bimanual touch and pen input on and above a tablet as well as gaze [11]. The application includes hybrid bimanual interactions, but the bare hand is similarly confined to a minimal role of activating quasimodes and basic panning and zooming operations with touch while the pen performs most of the work and eye gaze is used for object selection.

## 3 PEN+TOUCH+MIDAIR DESIGN SPACE

To present our design space, we follow established practices of identifying the dimensions and core building blocks of the space, situating prior work within the defined taxonomy, and giving further examples of how interaction techniques can be constructed using the proposed framework [18,19,29,48,67].

Our design space is inspired by previous research on bimanual pen and touch input on tabletops and tablets. A fundamental principle that underlies much of this research is Guiard's kinematic chain model for asymmetric division of labour, which establishes that the non-dominant hand—or "other hand"—creates a frame of reference within which the dominant hand operates [26]. Based on that model, initial explorations of two-hand hybrid pen and touch

input [13,36] reveal how each modality can efficiently complement each other, either as a succession of smooth "phrases" of interleaved pen and touch operations (e.g. the other hand pans and rotates a canvas and the hand holding the pen writes on it) or combined input, which creates "new tools" (e.g. pinning a GUI element with a finger of the other hand and dragging it away with the pen to create a copy) [36]. This paradigm of complementary and synergistic gestures has been applied in a variety of contexts, including graph editing [23], math [83], document authoring [52], gaming [30], and sketching [34,61].

### 3.1 Design Space Dimensions

We seek to extend the well-studied pen+touch input space on the surface to include interaction *above* the surface in XR. As described by Marquardt et al. [48], these spaces on and above the surface can be considered a "continuous interaction space", where interactions fluidly move from touch to midair and vice versa. We can therefore derive two primary dimensions for a design space combining pen + barehand input across this interaction continuum: the *hand* used for the interaction (the dominant *pen hand* and the non-dominant *other hand*) and the interaction *space* (2D touch *surface* and 3D *midair*) (Figure 2). This results in four possible asymmetrical two-hand combinations forming hybrid interactions: two within the same space and two across spaces. Furthermore, within the *hand* dimension, interactions that incorporate cross-space transitions and combinations can be defined from surface to air and vice versa.

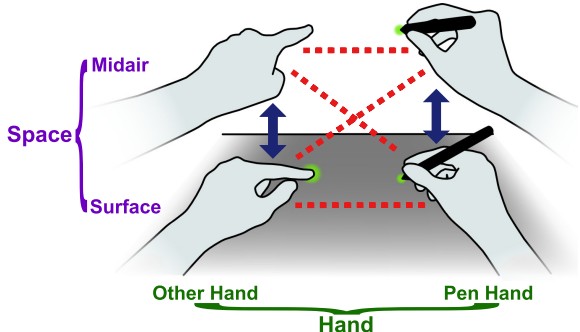

Figure 2: Design space of asymmetric bimanual pen and barehand input in XR: red dashed lines are combinations between hands and interaction spaces; dark blue arrows are cross-space transitions.

### 3.2 Hand and Input Space Characteristics

Table 1. Hand and input space characteristics with examples of suitable interactions for each combination

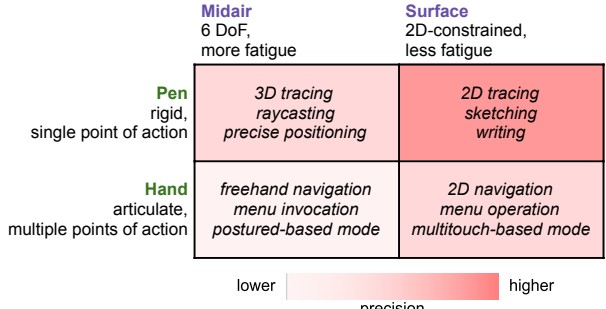

| | Midair 6 DoF, more fatigue | Surface 2D-constrained, less fatigue |
|---|---|---|
| **Pen** rigid, single point of action | *3D tracing raycasting precise positioning* | *2D tracing sketching writing* |
| **Hand** articulate, multiple points of action | *freehand navigation menu invocation postured-based mode* | *2D navigation menu operation multitouch-based mode* |

lower ⟶ higher
precision

The pen and the other hand as well as each input space individually have different properties. With its pointy tip, the pen is a more precise instrument than the bare hand and its fingers, both on a surface [51] and in midair [1]. However, while the pen only typically supports single-point input, the articulate hand and its fingers can perform a wider range of interactions, including multitouch and forming postures. As shown by prior work on tablet + midair input, the

hard surface of the device provides support for constrained and precise 2D input, while midair 3D interaction is unconstrained but coarser [4]. The support afforded by the surface additionally reduces fatigue (arms and wrists can rest on the tablet or desk) compared to midair input [32].

Table 1 summarises these characteristics and the relative precision afforded by each hand-space combination. Pen on surface provides the highest precision, but only supports 2D and single-point input, whereas barehand interaction in midair provides the most freedom and range, but is also the least precise and most fatiguing.

Another key difference between surface and midair interaction is that a physical touch surface provides haptic feedback to register input events even when there is no explicit interface element to "activate", for example when drawing or panning. In midair input, there is no such obvious signal for similar kinds of interactions [20]. Digital pens are often equipped with barrel buttons and therefore the equivalent of a touch, tap, or click in midair can just be a button press, much like pressing a button on a VR controller. For the bare hand, discrete triggers can consist in forming specific postures, such as pinching, making a fist, or a flat hand [74]. Our applications provide examples of posture-triggered actions and pen quasimodes.

As with non-XR systems, surface interaction can be used for both direct and indirect input to respectively control objects on and above the surface. We show examples of both in our applications.

### 3.3 Bimanual Interaction

In a classic pen and touch context on a 2D surface, each hand can interact independently, both hands can operate sequentially using interleaved pen and touch "phrases", and both hands can work together [36]. Simultaneous bimanual interaction patterns include the other hand defining simple constraints for pen input (e.g. virtual rulers and guides to precisely align traced content [24]) or activating "quasimodes", where pen function states are actively maintained by specific hand postures (e.g. four fingers touching the surface to enable a pen lasso selection mode [52]). We integrate midair space into this surface-based framework by examining how the other hand and pen can work independently, in sequence, or simultaneously within and across spaces. For instance, touch input from the other hand can set a mode for midair pen-tracing, or, in a 3D modelling application, the pen can extrude a 3D object by tapping a closed shape on the surface, then raising the pen to define the extrusion path in midair.

Both the pen and the palm of the hand holding it can touch the surface to perform unimanual pen+touch input for mode switching based on contact patterns [14]. Different parts of the pen can be associated with different actions, such as how some pen systems use the eraser end to enable a deletion mode. We include an example where the top end of the pen invokes a midair menu while the hand rests on the surface.

Another pen input dimension is rotation, such as rolling the pen barrel between the fingers to perform actions [10]. We also use this to create a "reset" gesture using both hands in midair.

### 3.4 Situating Relevant Prior Work

Table 2 situates previous non-XR and XR works that explore specific parts of our design space. As can be seen, outside of XR, there is extensive work on pen+touch and only one example of bimanual cross-space input with pen on the surface and the other hand in midair. In XR, prior systems have not examined simultaneous midair pen and hand input, nor have they considered cross-space midair hand with surface pen input. We address these gaps and investigate combinations between the two hands and interaction spaces (surface and midair) as well as unimanual cross-space transitions more systematically.

Table 2. Mapping previous bimanual pen+barehand desktop systems (XR and non-XR) into our design space.

| Application or Technique | Bimanual Same-Space: Surface Pen Surface Hand | Bimanual Same-Space: Midair Pen Midair Hand | Bimanual Cross-Space: Midair Pen Surface Hand | Bimanual Cross-Space: Midair Hand Surface Pen | Unimanual Cross-Space: Pen | Unimanual Cross-Space: Other Hand |
|---|---|---|---|---|---|---|
| *Non-XR examples* | | | | | | |
| Pen+Touch (e.g. [13,30,36,52,61,79,83]) | ■ | | | | | |
| Pen + Mid-Air [5] | | | | ■ | | |
| *XR examples* | | | | | | |
| Responsive Workbench [17] | ■ | | | | | |
| PoVRPoint [11] | ■ | | ■ | | ■ | |
| Pen Spreadsheet [25] | | | | | ■ | |

## 4 APPLICATIONS

We explore our design space through a set of interactions and manipulations in realistic contexts. These are implemented within three application scenarios: modelling, volumetric rendering, and terrain editing. These applications exploit the natural spatial reference frame provided by the stationary surface and require both coarse and precise manipulations, which pen and barehand input can offer. Full demonstrations are shown in the accompanying video and VR screen captures provided in the appendix.

While we aim for some degree of consistency and functionality across interactions, our demonstrations are intended as testbeds for hybrid interactions, not as optimised feature-complete applications. As such, we deliberately include some redundancy, including "mirrored gestures" (gestures performed on and above the surface that have the same effect) [48], to be able to investigate alternative methods for the same operation, such as moving objects, navigating in 3D space, or invoking menus in different ways. Our bimanual gestures follow Guiard's division of labour principle except for two gestures where the pen functions as a frame of reference for the other hand, thereby inversing the usual relationship.

### 4.1 Implementation Environment

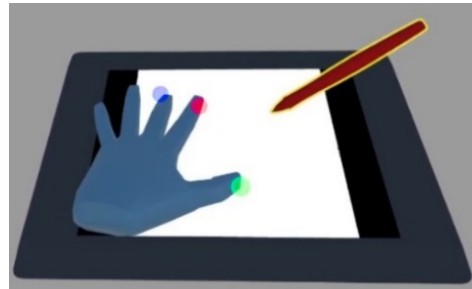

Figure 3: VR hand with touch points and pen.

In our system implementation, the user's other hand is materialised as a virtual hand model, but only the pen is rendered for the pen hand (Figure 3). When illustrating interactions in this paper, however, we also represent the hand holding the pen for better clarity. We use a pen and touch sensitive tablet for precise surface contact tracking. The tablet is placed on a desk, at which the user sits, and is simply referred to as "the touch surface" in the descriptions

below. Visual feedback shows the tablet surface with touch contact locations, and the virtual pen model "glows" when the pen button is pressed (pen contour outlined in bright yellow).

### 4.1.1 Apparatus

Our prototype applications are implemented in Unity running on a Windows PC with a GeForce GTX 1080 Ti GPU. Our system uses an HTC Vive Pro for the VR display and a Wacom Intuos Pro L with an active area of 311×216 mm for the pen and touch surface. An Optitrack motion capture system precisely tracks the headset, tablet, and pen in 3D. The required reflective markers are mounted on custom 3D printed mounts (see appendix for images). The Wacom system only reports pen button presses when it is in contact with the tablet. We therefore repurpose the internal hardware of a Bluetooth remote shutter as a barrel button, by attaching it with a resin sheath to the pen. The entire mount affixed to the pen adds respectively 27 g and 8.6 cm to its original weight of 18 g and length of 15.5 cm. Since it is not possible for our motion capture system to track non-rigid bodies like hands, we use a Leap Motion sensor fixed to the front of the headset to track the other hand. While it would have been desirable for the hand holding the pen to be tracked as well in order to be rendered in VR, the Leap Motion is not able to robustly track hands holding objects.

## 4.2 General Gestures

There are two interactions common to all our applications:

*Palm Rejection*: Palm rejection detects and discards unwanted touch input when the palm of the pen-holding hand contacts the surface. This can be relatively difficult in a classic pen and touch context, where only contact information from the touch sensors is available, but in our VR context, where we track both the pen and the other hand in midair, we can utilise that information to better determine which touch events should be rejected. We use the projected pen tip on the surface to discard touches occurring within a rectangle covering the whole hand and arm, and whose orientation aligns with the pen, a simplified version of Vogel et al.'s occlusion model [76]. Since the technique relies on first detecting the pen in midair followed by a palm touch, it can be considered a midair-to-surface transition.

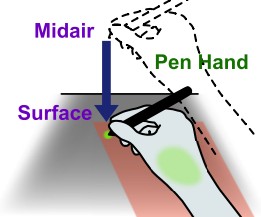

*Reset*: To our knowledge, no prior work on bimanual pen and barehand input has considered both hands manipulating the pen together and so we include one such interaction, where the pen is rubbed between the hands. Specifically, the palms are brought together in a "praying pose" with the pen clasped in-between. Then, the pen is spun back and forth by rapidly rubbing the hands together. Detection examines a 2-second window of accumulated pen rotations about its longitudinal axis, and the action is executed when a threshold is exceeded. To avoid false positives, this rotation threshold can only be achieved by rubbing with two hands. As this type of gesture is rather uncommon, we assign it to an action that is rarely performed but has a high penalty if accidentally triggered: deleting all objects and resetting the application.

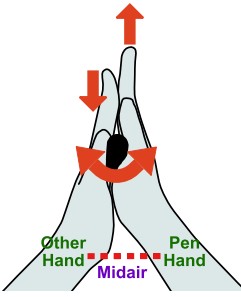

## 4.3 3D Modelling

Our first application considers simple 3D modelling where meshes are created by extruding shapes drawn on the surface with the pen

(Figure 1a). Our application is inspired by desk-based XR modelling systems like Mockup Builder [2] and DesignAR [64], but it also includes tools and gestures that fulfil various hand+space combinations in our design space. We create 10 types of bimanual and cross-space manipulations for this application.

*Drawing with Grid-Snapping:* Inspired by pen+touch techniques where pen input is constrained based on maintained touch postures of the other hand [13,24,36,52], we support grid-constrained shape drawing, where the grid is activated by touching the surface with two fingers of the other hand. The grid can be scaled using classic pinch/spread gestures. A third finger placed on the surface locks the grid so that small finger movements do not cause unwanted changes. While the other hand maintains the grid state, pen strokes snap to the grid, creating straight lines.

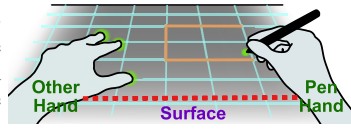

*Shape Extrusion with Freehand Path* and *3D Grid-Constrained Path*: Shapes are extruded by pressing the pen button to engage a midair operation, tapping inside a shape, and raising the pen without releasing the button. This pen transition from surface to air creates a volume from the base shape extruded along the midair pen trajectory when the button is released. Mockup Builder uses a similar procedure without an explicit touch inside the shape, which is less precise and explicit. A variation of this basic extrusion method uses the other hand contacting the surface to activate a context-based 3D grid [8] in order to constrain midair pen trajectory to straight line segments during extrusion. Thus, the other hand touching the surface sets drawing constraints for the pen both *on* and *above* the surface, with a smooth transition between the two spaces when performing extrusions.

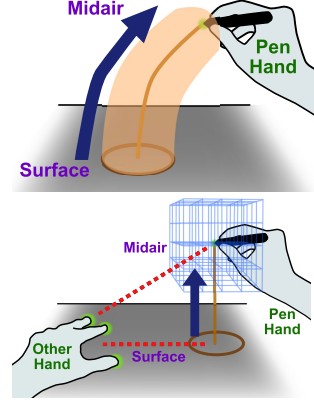

*Wrist Menu*: Wrist menus are common in XR applications using barehand interaction and they are a standard component of the Microsoft Mixed Reality Toolkit [55]. A typical invocation trigger used in applications is turning the palm up. However, using only that cue can lead to false positives. We include a menu summoned on palm up, but in our implementation, we also require the pen to be close to the wrist and point at the menu. Thus, this bimanual midair combination exploits the tracking and pointing precision of the pen to increase control over an action triggered by the other hand. Our menu consists of a colour palette and three buttons to select drawing and spraying tools with the pen.

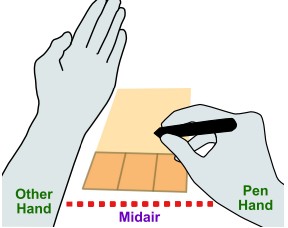

*Spray Painting Objects*: We include a spray-painting tool, which works by aiming the pen at an object and pressing the pen button. Contrary to existing VR graffiti and painting simulators, where the user paints on a fixed surface (unimanual input), we support spray-painting an object while it is held by the other hand, which allows the user to continuously adjust the position and orientation of the object in order to paint different sides. This

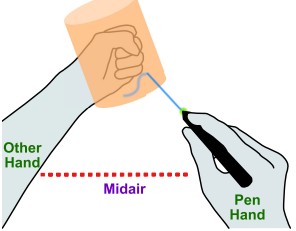

example of bimanual coordination between a pen (virtually) inking a constantly adjusted surface is the 3D equivalent of the pen-on-paper example used by Guiard to illustrate how the sheet of paper functions as the frame of reference that is repeatedly repositioned by the non-dominant hand for the writing task performed by the pen held in the dominant hand.

*Object Delete*: Similar to mode switching triggered by contact patterns on the touch surface, midair hand postures can also be defined to change the pen mode [5]. As an example of such a posture, we choose an "ok-sign", formed by an index-thumb pinch with raised middle, ring, and little fingers, which is a familiar gesture that can

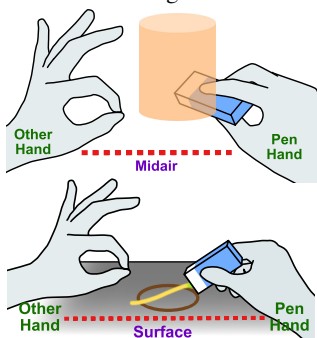

be robustly detected with almost no false positives. When the posture is active, midair objects can be deleted by placing the tip of the pen inside the object and pressing the pen button, while surface shapes are deleted by crossing them out with the pen. Therefore, like the two-finger grid-invoking touch posture, this mode-switching pose enables the same action both on and above the surface.

*Shape/Volume Duplication in 2D and in* 3D: An example of a pen+touch gesture used in several prior works is a pin-copy-drag action, where a 2D shape is pinned by a finger of the other hand and a copy created by dragging away with the pen [36,52]. We support this gesture and create an analogous midair gesture for 3D objects, where a finger of the other hand is inserted into a volume, following which the pen can point inside the object and pull away to create a copy.

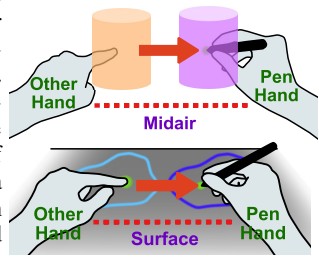

*Volume Stamping:* When characterising the continuous interaction space, Marquardt et al. give several examples of gestures starting on the surface and extending above [48], but do not consider the opposite direction. We provide an example of a transition from midair to the surface, where 2D stamps of a 3D volume can be created by grabbing a volume with the other hand and touching the surface while the volume intersects it. The contour of the cross section formed by the intersection of the volume and the surface creates a new 2D shape that can in turn be used as a base to extrude other volumes.

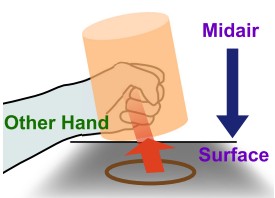

## 4.4  Volumetric Rendering

Our second application is for interactively exploring volumetric data, where 2D cross sections of a 3D volume can be isolated for inspection and annotation. A typical example of volume datasets where cross-sectional slices are used is medical data, such as computed tomography (CT) scans. When exploring data using interactive systems, cross sections or slices can be defined via multitouch input and hand orientation [70] or by moving a physical artefact through the volume, such as a piece of cardboard [72] or a tracked mobile device [9,58,68]. Following Guiard's division of labour for the two hands, we can assign the role of slicing through the volume data to the other hand (setting the frame of reference) and annotating target frames to the pen (main task). Concretely, in our example scenario, the bare hand is used to form the plane that cuts through

a floating 3D skull to select slices of virtual X-ray images. These slices can then be annotated with the pen on the touch surface (Figure 1b). This combination of spatial selection in midair and annotation on the hard surface creates a natural cross-space environment that is conducive to exploring our design space. Our application includes four bimanual and cross-space interactions.

*Setting and Annotating Slices*: Slicing is performed by moving the flat other hand inside the floating virtual skull. The current slice corresponding to the hand position is projected on the touch surface as well as on a vertical panel to facilitate visualisation from different angles (Figure 1b). Since the skull can be freely grabbed, moved, and rotated, its orientation can be easily adjusted to comfortably achieve desired slice angles. To be able to annotate a slice with the pen on the surface, it first needs to be locked. We support two locking techniques corresponding to two use cases. The first method is quasimodal and consists in flexing the thumb into the

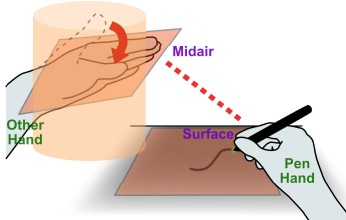

palm (see illustration). This allows to rapidly engage and disengage the lock for quick successive annotations while probing the volume with the other hand. Since this is a maintained mode, the hand needs to remain in the volume while locking/annotating (bimanual cross-space interaction). Midair-activated quasimode for pen input on the surface are unconventional (Aslan et al. only considered quick gestures [5]) and we include this type of cross-space combination to investigate potential usability and fatigue issues in our study. For comparison, we also include a second locking method, which is a more conventional mode switch activated by tapping the surface around the projected slice with the pen. In our implementation those activation zones are two large rectangular regions on both sides of the image. Compared to the maintained mode, the cross-space coordination of the two hands in this case is short as it is only required when selecting the slice with the other hand and tapping the surface with the pen. After the slice is locked, the other hand can be removed from the volume so that it can rest on the surface, making the technique appropriate for longer, more detailed annotation in a more comfortable pose.

*Pen Tilt Menu*: Single-hand cross-space interactions need not involve an explicit transition from one space to the other. Gestures can be designed to require input in both spaces simultaneously, i.e. a touch contact together with a midair action. This type of hybrid interaction in the continuous interaction space [48], to our knowledge, has not been explored in

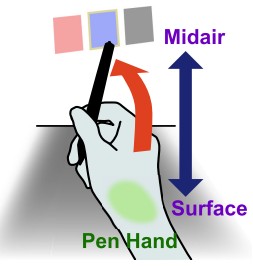

prior work, let alone for pen and touch. We propose a novel gesture fulfilling that condition to invoke a colour menu. It is performed by resting the hand on the surface (i.e. touch input) like when writing or sketching normally and tilting the top of the pen forward. When a threshold tilt angle is exceeded, a menu appears in the vicinity of the pen top. A menu item can then be selected with a midair tap. Requiring both a palm touch and a pen forward tilt makes sure the menu is not accidentally triggered in midair when the hand is more likely to hold the pen in different orientations. Identification of the touch point(s) as originating from the pen hand is handled the same way as for palm rejection and therefore it does not interfere with touch input from the other hand. Due to the limited range of pen movement when the palm of the hand is resting on the surface, only a small number of menu items can be supported with this technique.

*Slice Selection and Copy*: Slices with associated annotations persist inside the skull. These slices can be selected with the other hand or the pen for viewing on the surface. Furthermore, copies can be moved to other areas of the virtual space for comfortable side-by-side comparisons. These copies can be created in two ways: By pressing the pen button while the pen is inside a slice and dragging the copy away. Or, by selecting the slice with the other hand and pressing the pen button at the desired target location of the copy. This bimanual copy method minimises pen movement, as it allows the other hand to focus on selection, while the pen remains in the area where copies are placed. This is also an alternative to the copy method of the first application, where the pen must be inside the selected object to initiate the copy.

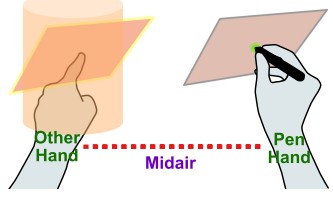

*Pen-Constrained Rotation Axis*: On the responsive workbench [17], Cutler et al. showed how, in a Guiard-abiding way, the other hand can form poses to constrain volume rotations performed by the pen. However, the pen is a long and thin instrument that we think is very fitting for setting axes and line-based constraints. We therefore choose to use the pen as the constraint-setting instrument for controlled rotations around specific axes, thereby breaking from Guiard since the pen both precedes the other hand as well as sets the frame of reference. To set a rotation axis, the pen is placed inside the volume (the axis is shown as an infinite blue line). Grabbing the skull with the other hand then rotates the object around that axis. The pen can be moved to dynamically adjust the axis while rotating with the other hand.

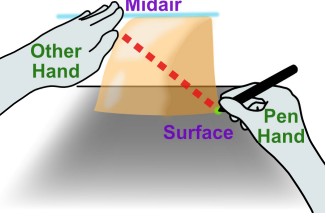

## 4.5 Terrain Editing

Lopes et al. showed how multitouch on a stereoscopic tabletop display can be used in conjunction with a wand in midair for 3D geo-modelling [47]. We extend and modernise that approach to create a terrain editor in full VR (Figure 1c) operated by hybrid gestures addressing the four hand and space combinations of our design space. We create these gestures to explore different methods to edit and navigate in the landscape as well as invoke menus. There are four main bimanual interactions.

*Terrain Sculpting*: Similar to Lopes et al., terrain can be sculpted and features such as trees can be added by pressing the pen button and waving it in midair, where the height of the inserted landscape is determined by the 3D position of the pen. Those editing operations can also be performed directly on the surface for more precision. In that case, terrain altitude is determined by the height of the other hand above the surface, as shown in the illustration.

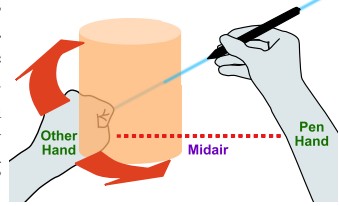

*Navigation*: When using touch input, the user can move the camera to navigate through the terrain by dragging with fingers on the surface like using a trackpad. One finger pans, two rotate, and three change the elevation (dragging upwards/downwards on the surface moves the camera up/down). Alternatively, navigation can be performed with the other hand in midair by forming a grabbing pose and moving the fist. The camera can be iteratively moved with repeated grabbing and pulling, similarly to Coomer et al.'s point-tugging technique [16], but using the bare hand instead of a controller.

Since DoF are separated when navigating with touch while the other hand in midair controls all DoF at once, we add the possibility to constrain midair movements to panning only by extending the thumb.

*Editing While Navigating*: As shown by bimanual pen+touch techniques like Bi-3D [61], 3D sketching can also be performed with the other hand responsible for the main tracing motion while the pen remains mostly stable (another departure from Guiard). In a similar way we support sculpting while moving the camera with the other hand using any combination of the editing and navigation methods, i.e. maintaining the pen button pressed in midair or contacting the surface with the pen while dragging on the surface or grabbing-pulling in midair with the other hand. These flexible techniques allow rapidly adding terrain elements over large areas, e.g. a long ridge or line of trees (see accompanying video). We believe this novel way to sculpt landscapes in VR is a particularly powerful example of barehand + pen input, which we surmise would be more difficult to achieve with two controllers.

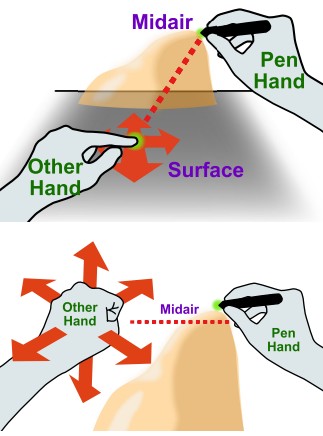

*Marking Menu*: In the first application, we considered wrist-anchored menus, but VR menus can also be bound to world space. We include a marking menu to change the current terrain pattern. Here as well we support several invocation techniques. One method requires touching the surface with four fingers of the other hand to display the menu in the middle of the screen, then dragging towards the desired item and releasing the fingers to select (unimanual technique). The cursor is anchored to the rightmost finger (if the other hand is the left hand) so that gesture relaxation [80] is possible by dragging only with that finger after triggering the menu. A second bimanual method (which requires a manual switch to be enabled) consists of a trigger from the other hand followed by pen marking. Invocation starts similarly with a four-finger touch of the other hand, but the menu appears at the location of the pen tip (see illustration). The pen is then moved inside the desired menu item and the fingers of the other hand released to confirm the selection. Alternatively, the menu can be invoked by turning the palm of the other hand up in midair, like the wrist menu of the first application. Together, these three techniques constitute

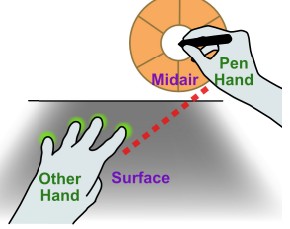

novel ways of invoking and using menus in VR, as they differ from previously proposed bimanual marking menus for multitouch interfaces, which were mostly designed to support multi-level menus [40] and for higher integration with tracing tasks [27,60].

## 4.6 Design Space Coverage

Using the same structure as Table 2, Table 3 maps the 23 different spatial combinations and transitions used for interactions in our example applications. These broadly cover the pen+touch+midair design space and fill in the gaps not addressed by prior work in XR. We provided two examples for each unimanual cross-space interaction, at least three examples for bimanual cross-space combinations, nine examples of mid-air pen+barehand interactions (which prior work had not considered at all), and several instances of input overloading for the same operation to directly compare alternative

techniques. While, of course, we have not exhaustively explored the design space, we hope the many diverse examples we have given will inspire designers of future pen-based desktop VR systems.

Table 3. Mapping the example interactions used in our three applications into our design space.

| Application or Technique | Bimanual Same-Space — Surface Pen / Surface Hand | Bimanual Cross-Space — Midair Pen / Midair Hand | Bimanual Cross-Space — Midair Pen / Surface Hand | Bimanual Cross-Space — Midair Hand / Surface Pen | Unimanual Cross-Space — Pen | Unimanual Cross-Space — Other Hand |
|---|---|---|---|---|---|---|
| **All Applications** | | | | | | |
| Palm rejection | | | | | | ■ |
| Reset | | ■ | | | | |
| **3D Modelling** | | | | | | |
| Drawing with grid | ■ | | | | | |
| Shape extrusion | | | ■ | | ■ | |
| Wrist-anchored menu | | ■ | | | | |
| Spray painting objects | | ■ | | | | |
| Object delete | | ■ | | ■ | | |
| Shape duplication | ■ | ■ | | | | |
| Volume stamping | | | | | | ■ |
| **Volumetric Rendering** | | | | | | |
| Locked slice annotation | | | | ■ | | |
| Pen tilt colour menu | | | | | ■ | |
| Slice selection and copy | | ■ | | | | |
| Pen-constrained rotation | | ■ | | | | |
| **Terrain Editing** | | | | | | |
| Height-controlled editing | | | | ■ | | |
| Editing while navigating | ■ | ■ | ■ | ■ | | |
| Marking menu | | ■ | ■ | | | |

## 5 USER EVALUATION

We conducted a qualitative user study to assess the usability and suitability of our techniques. We sought feedback from participants to form a broader analysis and discussion on practical aspects of our design space. Based on that feedback we further hoped to identify issues of interest at a more granular level for future focused investigations. Questions that we were particularly interested in are: Can people efficiently coordinate two hands in two input spaces? How intuitive and complementary are bimanual cross-space interactions? What conditions might influence preference for manipulations with the pen vs the other hand, surface vs midair?

The study design was a walkthrough-style task, where participants tried the three applications and their interactions and provided oral feedback. We recruited 16 volunteers from within our institution: 13 males, 3 females, with mean age 38 (SD=6.9). Nine had previous experience with VR, with seven people owning a VR device. None had ever used a pen in VR or for midair input.

### 5.1 Procedure

Before starting the study, the participant adjusted the height of their chair and the desk to their liking. We then demonstrated the two delete gestures (delete all/reset and single delete with ok sign) and the tilt menu as they were more difficult to explain once the

participant was wearing the headset. Next, the participant donned and adjusted the VR headset so that it fit comfortably and the display was clear and focused. We then explained the gestures and techniques in the first application one by one, each time asking the participant to try them. After explaining all the gestures, the participant could freely "play around" to further familiarise themselves with the interactions. This pattern was repeated for the other two applications.

At all stages, the participant was encouraged to share their impressions, but a short interview after experiencing each application formed the primary source for the collected feedback. During the interview, we asked for specific preferences between the different alternative methods they experienced in each application. In all cases, we solicited comments focused on the gestures rather than functionality and minor UI design concerns. We further asked them to ignore tracking issues as much as possible and assume hand detection was robust when judging the techniques. Each session took approximately one hour. A selection of snacks was offered to participants as a thank-you.

### 5.2 Results

Participant feedback was compiled from detailed notes and similar feedback was grouped by the lead author to form the basis of the analysis. We first report on general usability observations, followed by feedback on the individual applications and their techniques.

#### 5.2.1 General Observations

Overall, participants did not encounter major difficulties when performing the gestures, even though most were unfamiliar to them. One issue that initially confused four participants was when to press the pen button to initiate a pen action. When using the pen on the tablet, surface contact is the inking and action trigger, whereas in midair the button needs to be pressed. One participant said that if the pen is used both on and above the surface, they would prefer to consistently use the button all the time for any pen action.

Participant feedback generally confirmed that manipulations on the tablet were more precise than in midair, but more limited in space and slower. Similarly, the pen offered more precision than the bare hand, but lacked the expressiveness of an articulate hand.

The applications include different techniques, where both hands were used on and above the surface, creating opportunities for some rest when using them on the surface. Like Aslan et al., we observed that for low-height midair manipulations, some participants would rest their elbow on the surface to reduce arm effort [5]. Nevertheless, hand and arm fatigue from midair use was an issue for three people, mainly for the other hand, which was perceived as doing more work than the pen-holding hand. This is interesting because the pen is an additional weight compared to the free bare hand. Two participants said the pen was slightly heavy due to the mount.

In general, participants valued the possibility of using postures with the other hand to set quasimodes as an alternative to selecting with menus. This was true for surface and midair postures, although midair quasimode postures were considered more fatiguing if maintained for a long time.

Regarding the *Reset* technique, opinions were evenly split. Half of the participants said the gesture was easy to remember and a good choice for this action as it was difficult to trigger unintentionally. Yet the other half of the participants expressed difficulty when rapidly spinning the pen between their hands because of the thick resin attached on one side of the barrel. This problem is a side-effect of our prototype pen button and would likely not occur with a perfectly round pen.

#### 5.2.2 Modelling Application Interactions

*Shape Extrusion with Freehand Path* was considered easy and intuitive by 13 participants and *Extrusion with 3D Grid-Constrained*

*Path* was deemed practical by 12 participants, as it was similar to maintaining a modifier key pressed on a keyboard. All however agreed that using the 3D grid was more difficult. Locking the grid using a third finger was only acceptable for 9 participants, the others feeling the sequence of action needed was too convoluted. The *Wrist Menu*, invoked by turning the palm up, was considered an effective gesture by all but one participant, confirming the popularity of that technique and its inclusion in many XR applications. The participant who disliked it said they preferred to use menus in a fixed screen location rather than tied to the wrist.

Interactions based on physical metaphors such as *Spray Painting* and *Volume Stamping* posed no major problems, even though hand tracking issues sometimes made it difficult to grab and orient volumes as desired. For *Volume Stamping*, four participants said it was sometimes not easy to obtain a desired intersection shape upon contact with the surface. Activating the *Object Delete* mode with the "ok sign" posture was found suitable by 12 participants, but two noted it did not have any strong semantic association with deletion (unlike a throwing motion, which was suggested by one of those participants). 13 participants found the *Shape/Volume Duplication* gestures to work well both on and above the surface, which hints at the portability of pen and touch gestures to the midair space.

### 5.2.3 Volumetric Renderer Interactions

Even though it does not follow Guiard's kinematic chain framework, the *Pen-Constrained Rotation Axis* technique was considered a compelling metaphor and easy to perform by 14 participants. 12 participants found the *Pen Tilt Menu* easy to invoke and useful, with two participants finding they could use the knuckle of their pinkie as pivot on the surface to comfortably tilt the pen forward while satisfying the touch condition. The four people who had some difficulties with the menu said the tilting motion felt unnatural and that there was a risk of grazing the surface with the pen nib.

12 participants found *Setting and Annotating Slices* inside the virtual skull using a flat hand an intuitive way to position and orient a plane in 3D space, with three people saying they would also like to complement that gesture with 3D widgets or multitouch gestures for precise positioning. For locking, participants overwhelmingly preferred doing so with a pen tap (13), but five participants admitted locking with the thumb allowed them to focus on a single hand during the operation, and while it was more demanding to maintain this posture in midair, they agreed it could be useful for successive short annotations (which was the main intention of this gesture).

When asked if they preferred pen or barehand "grabbing" for *Slice Selection and Copy*, the pen was said to be more precise (six people) but grabbing with the other hand felt more natural (two people). Five participants said they liked both options, since the decision to grasp with the pen hand or the other hand may just depend on which is closer. Bimanual copying (index finger inside a slice and pen button to create a copy) was only preferred by two people. The rest thought using only the pen to both select and copy was easier even though it required more hand movement. We further observed participants rearranging copied slices in the 3D space using both hands simultaneously, meaning that the type of input device or limb does not necessarily dictate a specific usage.

### 5.2.4 Volumetric Renderer Interactions

Participants enjoyed the terrain editor the most as it was playful and its potential use for game design and world building "à la Minecraft" was appealing. People found waving the pen in midair for *Terrain Sculpting* natural and fun. *Navigation* with multitouch drag gestures felt like using a large trackpad and therefore was the preferred locomotion method for 13 participants. The DoF separation further allowed movements to be more controlled and, when combined with pen editing (*Editing While Navigating*), supported long-distance but precise landscaping. The other five people preferred midair grabbing, as they found they had more freedom to position the hand and move it anywhere, compared to the tablet, where input was only possible in a limited surface area. Midair grabbing allowed faster navigation, but quick movements also caused motion sickness for four participants and repeated clutching was fatiguing. Furthermore, two participants said that grab mode was more likely to be accidentally detected than touch. Three participants stated they liked to have both navigation options, depending on how fast or carefully they would like to move in the virtual space.

Regarding the *Marking Menu*, seven participants preferred invoking and marking with four fingers on the surface, the easiness and convenience of using only one hand given as the main reason. Five people favoured the bimanual techniques, mainly because the pen allowed more agile and precise selection. Four participants preferred the midair palm up gesture to invoke the menu compared to one favouring the four-finger tap, showing again the strong intuitiveness of that first gesture. Two participants said they would like to use touch with the other hand only for navigation to clearly separate hand and input space roles. The remaining four people found all three methods equally acceptable.

Editing the terrain with the pen on the tablet was considered precise to add detailed features like trees (five participants). Four participants stated they preferred to use midair editing only if both height and ground position needed to be controlled.

## 6 DISCUSSION

The qualitative evaluation format enabled us to cover a range of usability aspects relevant to our design space with an emphasis on bimanual same-space and cross-space interactions in different application contexts. We summarise overall trends and emerging characteristics of this space. Our motivation is not to make definitive conclusions, but to show general promising directions and to motivate future more focused studies and dedicated experiments.

*1. Bimanual interactions within the same space are perhaps the most straightforward, since both hands share the same spatial constraints.* Bimanual surface gestures can be used for precise input, whether it is to trace with the pen, or use a finger touch to define a grid or navigate. Two-handed midair gestures benefit from a larger interaction space and more DoFs can be controlled simultaneously, but input is coarser and more fatiguing.

*2. The combination of both spaces offers the most flexibility, with the possibility to perform manipulations at different levels of granularity and with different spatial constraints.* Our terrain editor, which supports navigation and editing in both input spaces was a good testbed for this trade-off. The asymmetric nature of pen and barehand input further expands that flexibility with the pen responsible for more precise operations than the bare hand. Our design space enables a meaningful comparison of fatigue, precision, and degrees of freedom across space and hand dimensions (Table 1).

*3. Cross-space gestures with the pen in midair and the other hand on the surface seem to be more practical than gestures combining pen-on-surface with the other hand in midair.* The use of the other hand for touch-based navigation or mode-setting while the pen sketches or edits in midair appeared to be particularly effective. Terrain editing, where navigation over large virtual spaces is required, seemed like a particularly compelling use case for this interaction pattern. Gestures combining touch input with one hand and the pen in midair are still very much underexplored, and we see this type of cross-space hybrid input as particularly promising and worthy of future research, including in mobile contexts using other pairings like phone+pen. The opposite combination, pen on surface+other hand in midair, however, seems to have more limited use, as evidenced by the feedback we received for our thumb-locking technique for volumetric slices and height control for terrains. The requirement for the other hand to remain in midair to maintain

a surface mode or set a parameter is probably better kept for short switches or simple actions (as also advocated by Aslan et al. [5]) like our hand posture for quick deleting. The effort imbalance seems to be more perceptible compared to bimanual midair gestures, because one hand is resting on the surface while the other is in midair. While we did not ensure that both hands spent equal time in each space, participant feedback suggests that the other hand is more prone to fatigue than the pen hand in these contexts. This might be because the pen is more precise and therefore movements of the pen hand can be more economical compared to barehand input, which is rougher and therefore more physically demanding. This explanation is supported by the fact participants preferred picking and moving cross-section slices with the pen rather than the other hand. For midair objects at low height, however, users can reach for them while resting their elbows on the surface, thereby reducing arm fatigue. Designers of desktop XR applications may therefore want to consider keeping interactive objects above the surface within forearm distance.

*4. The most compelling use for unimanual cross-space input may be in implicit cases like palm rejection.* For explicit transitions, cross-space gestures should likely be limited to transfer actions such as picking and placing objects or transforming them from 2D to 3D and vice versa [48]. With the pen, cross-space input may further cause some initial confusion about when to use the barrel button to trigger pen actions.

*5. Defining maintained modes for same-space interaction as well as cross-space surface hand+midair pen is sensible, but we recommend against those modes for cross-space midair hand+surface pen.* In terms of quasimodes, our observations suggest the fatigue imbalance between the two hands is more perceptible with cross-space interactions, with surface hand+midair pen seemingly being the more tiring. For same-space interactions, however, there was equal acceptance of bimanual techniques, not only those that mimic real-world interactions (like painting on or spinning an object with one hand while holding it with the other), but also more abstract postures for mode-switching (e.g. 2D and 3D duplication gesture). This also supports the idea of mirrored gestures [48].

*6. The wrist menu seems to be the favoured option if the other hand mostly operates in midair, but menus triggered and controlled by touch are also practical, as they are quick, efficient and can reduce fatigue.* Participants' slight preference for invoking the marking menu using touch even when the menu appears above the surface confirms the viability of indirect touch input for VR menus. We did not include classic touch-operated UIs displayed directly on the surface, but of course such kinds of interfaces are also possible. However, if most content is above the surface, the user would be required to frequently look down, which may not be desirable.

*7. Single-hand cross-space input has promising potential.* In addition to menu invocations using the other hand, we proposed a novel technique to show a menu with a forward pen tilt and item selection with a midair pen-top tap. Although this technique may require a little practice for some people, it may be an option for users, who do not want to interact bimanually. Unimanual surface+midair gestures consisting of a touch base (a contact pattern) and a midair action performed by fingers of the same hand (e.g. a local context-based 3D menu) are possibly interesting interactions and we are unaware of prior work that has investigated their potential.

## 6.1 Possible Extension to AR

Our setting was virtual reality, but we believe our design space and findings are likely also applicable to other types of extended reality. In AR, for instance, the user's hands are visible so they do not need to be tracked for visual feedback if only touch and pen input are used. We think desktop AR is an ideal platform to combine pens

with other common input devices like keyboards, mice and mobile phones and maybe even regular handheld objects for cross-space hybrid bimanual interaction. If the surroundings are visible, pen+touch interaction on any available surface (i.e. not only horizontal) may further expand input possibilities. Ongoing efforts to create virtual or augmented workspaces for office work [54,71,77] may be good catalysts to explore this potential.

## 7 CONCLUSION

We presented a design space for desk-based pen+touch+midair interaction in VR featuring hybrid bimanual input on and above the surface. We outlined different possible combinations of the pen-holding hand with the bare other hand for same-space and cross-space interaction as well as unimanual transitions from one space to the other. We explored this design space in-context through modelling, volumetric rendering, and terrain editing applications which include several techniques and gestures exemplifying these combinations. We evaluated their usability and suitability in a qualitative study with 16 participants. Their feedback highlighted how the two spaces are complementary since they flexibly support precise constrained 2D input on the surface and fast unconstrained 3D interaction in midair with both hands. Among cross-space combinations, we find touch with the other hand + midair pen to be a very promising synergy that strikes good compromises between precision, freehand input and fatigue. Understanding future interactions for practical desk-based work remains a challenging goal, but we believe combining touch and pen input on a physical surface with the potential of midair 3D input greatly expands the possibilities.

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
