# OpenReview forum: "Pen+Touch+Midair: Cross-Space Hybrid Bimanual Interaction on Horizontal Surfaces in Virtual Reality"
_graphicsinterface.org/Graphics_Interface/2023/Conference — GI 2023_

### Official Review · Reviewer_Xbv3 · 2023-01-05
**Good exploration, interesting ideas, nice that there is an observational study**

**Rating:** 7
**Confidence:** 4

**Review:**

This submission explores the design space of surface, mid-air, and cross-space interaction when using pen and touch in VR. It describes the design space itself, situates prior work within it, details examples of gestures that fit within the design space using three applications, and presents the findings from a walkthrough suer study that utilized the proposed gestures. The contribution of the submission comes from the design space, proposed gestures, and study findings. This submission fits within the scope of GI.

Overall, I enjoyed this paper. It is well written and explores an important aspect of pen-based interaction (yes within the context of VR, but many of the ideas could also apply to AR as well). I especially appreciate the reset praying pose gesture – sort of like pen rolling but using two hands. I also liked the spray painting objects and stamping gestures / interactions – going towards the surface isn’t well explored and the hand as a proxy with the pen was nice to see. The pen tilt color menu was also a good idea – I don’t particularly like the mapping of this gesture to a color menu because it made me think that the vibrancy or brightness of the colors would be mapped to the pen orientation level, but nevertheless, I could see this being used for other mappings.

I guess my major criticism of the submission is that 4.6 was basically a single sentence and table without any exposition on, perhaps, how difficult it is to design cross-space gestures or other hand gestures. As the related work is really long (and the paper is long in general), it might have been better to shorten the related work and include at least a paragraph discussing the challenges / opportunities encountered in the design process. It would also be useful, where appropriate, to include references to the prior work that inspired the gesture / interaction, as, by the end of Section 4, it was difficult to understand which gestures were truly novel versus adaptations of existing proposals.

I also liked the use of the walkthrough study – this is more of a design paper, so conducting a stringent quantitative study doesn’t really fit. I appreciated that several people used the applications a provided feedback. It would have been nice to have actual participant quotes. If they exist, it would be useful to include a few. One other note, in 5.2, it was difficult to map the gesture / interaction that was discussed to the names used in Section 4. Maybe using bolding or italics and the same capitalizations and phrases to make this clear? Or give each gesture/interaction and letter or number in 4 and use it in 5.2? I found it difficult to get through 5.2 as written because there are so many gestures to keep track of in your head.

Lastly, I wasn’t particularly excited about 6.1 – this seems like something that should have been integrated within Section 3? It doesn’t describe knowledge that arose from the observational study (or if it does the tie is really unclear), and it seems a better thematic / contextual match for the design space section.

In summary, I think this paper would be a good fit for GI. It presents a nice exploration of these design space dimensions and the applications themselves led to some creative functionality that, while not generalizable to a every other application, was interesting to read and seems plausible within other VR and AR contexts.

Other:
-	Et al -> et al.
-	Add spaces between the numbers in the references to make reference chains easier to read (e.g., [3,5,35,46] -> [3, 5, 35, 46])

---

### Official Review · Reviewer_Mnuh · 2023-01-13
**Clearly written article but I would expect some more clarification.**

**Rating:** 7
**Confidence:** 4

**Review:**

In this manuscript, the authors explored a design space for touch and pen bi-manually interaction in VR space. The authors also gave three examples of designing interactions techniques in cooperating with this design space as we;; as an evaluation.

I think the topic explored is interesting and useful. It fits the scope of this conference. Below are my comments:

The authors have clearly stated that the design space exploration was based on the desktop-VR metaphor both in the abstract and the supplementary video of this manuscript. However, this concept was not well explained in the introduction (and other parts of the main text), thus it is a bit confusing at the beginning since VR+large could mean many different setups. I would suggest the authors explicitly describe the setup they based on, especially how the 2D surface was placed and aimed for inside the VR view, even though the design space could be extended to other setups.

 I appreciate the large number of related works the authors have included in this manuscript, however, I think the narrative was not focused. For example, there are only brief summaries in sec 2,1. it is not clear how exactly these interactions were used and whether they can be transferred/used in VR.

  The authors omitted discussions of the display space: like what might be visualized on the 2D surface and what in the 3D space. While it is not wrong to focus solely on the interaction aspects, I personally think the display setup also affects the interaction, such as whether one interaction technique is direct or not. In addition, the three examples illustrated in this manuscript have quite some differences in visualization, I personally think adding some discussions with regard o this aspect would be useful (or a discussion of why this is not important here).

 While there are examples illustrating possible interactions, designing them seems not to be sufficiently related to the design space while proposing these interaction techniques. It would be useful for the community if the authors can provide guidelines or more concrete procedures on how to use this design space in each scenario. Because the valuable part of conducting a design space is not to show that a set of techniques could fall in part of it, but to show that it can guide the future design.

For the evaluation part, I expect some explanations on why evaluating these techniques is a good reflect to the evaluation of the design space.

To sum up, although I have mentioned a few concerns and expect more explanations, I overall like the idea of this design space and do believe that it could be quite useful and suits the scope of the conference.

---

### Official Review · Reviewer_ER7g · 2023-01-14
**Paper review**

**Rating:** 8
**Confidence:** 4

**Review:**

This paper presents the exploration of the design space for hybrid bimanual touch and pen interaction extended to midair interaction in desktop VR.  The paper presents the design space and uses 3 prototype applications to explore different interaction techniques that fall within the space.  The 3 prototypes include modeling, volume rendering, and terrain generation.  The paper also presents a small user evaluation that exposed 16 participants to the 3 prototypes which provided insights into navigation, object manipulation, and menu invocation.  The results from the study are used to present design implications.

Overall, this paper presents interesting research that is timely and will be a valuable tool for other researchers who want to explore this space.  The paper is well written and easy to follow. It presents a significant contribution to the field.  The references are good although the following paper should be included

Forsberg, A., LaViola, J., and Zeleznik, R. "ErgoDesk: A Framework for Two and Three Dimensional Interaction at the ActiveDesk", In the Proceedings of the Second International Immersive Projection Technology Workshop, Ames, Iowa, May 11-12, 1998.

One area where the paper could be improved is with the descriptions of the different interaction techniques.  The "what" of the techniques is presented, but the "why" is missing.  Why did the authors choose those particular pen, touch, and midair combinations?  It would be good to provide more clarity on the design choices made.

---

### Meta-Review · Area_Chair_hJXX · 2023-01-19

**Recommendation:** 8
**Confidence:** 4

**Metareview:**

All three reviewers agreed that the topic presented is original, important to the fields, and of high relevance to the GI conference. They also agreed that the clarity of the presentation is overall nice, with only a few minor suggestions as mentioned by each reviewer. In addition, the three applications (with the supplementary video) demonstrated the example use cases of the theoretical design space, which is also a pro of this submission.

Thus, the score of this metareview is based on the average of all scores, taking the importance of suggestions given in reviews into account. Overall, all improvements expected are not critical to judge the contribution of this work, which explains the positive final score.

However, we still expect the authors to carefully read all three reviews and try to revise accordingly as much as possible.  In particular, the list of comments below is that the authors may primarily consider revising since they are very related to the core and usefulness of the proposed design space.
 - Explain the design reason for choosing these particular interaction techniques.
 - Explain how users could make use of the design space.
 - Make references to prior work and explain how they inspired the design.